# Identification of Flavanone 3-Hydroxylase Gene Family in Strawberry and Expression Analysis of Fruit at Different Coloring Stages

**DOI:** 10.3390/ijms242316807

**Published:** 2023-11-27

**Authors:** Yanqi Zhang, Yongqing Feng, Shangwen Yang, Huilan Qiao, Aiyuan Wu, Jinghua Yang, Zonghuan Ma

**Affiliations:** College of Horticulture, Gansu Agricultural University, Lanzhou 730070, China18893425315@163.com (Y.F.);

**Keywords:** strawberry, *F3H* gene family, anthocyanins, bioinformatics analysis, real-time fluorescence quantitative PCR

## Abstract

The color of strawberry fruit is an important appearance quality index that affects the marketability of fruit, and the content and type of anthocyanin are two of the main reasons for the formation of fruit color. At present, the research on anthocyanin synthesis mainly focuses on the phenylpropane metabolic pathway, and the *F3H* gene family is an important member of this metabolic pathway. Therefore, in order to clarify the role of flavanone 3-hydroxylase (*F3H*) in regulating anthocyanin accumulation in strawberry, we identified *F3H* gene family members in strawberry and analyzed their bioinformatics and expression at different fruit color stages. The results showed that the strawberry *F3H* family contains 126 members, which are distributed on seven chromosomes and can be divided into six subgroups. The promoter region of strawberry *F3H* gene family contains light response elements, abiotic stress response elements and hormone response elements. Intraspecic collinearity analysis showed that there were six pairs of collinearity of the *F3H* gene. Interspecific collinearity analysis showed that there were more collinearity relationships between strawberry and apple, grape and Arabidopsis, but less collinearity between strawberry and rice. Via tissue-specific expression analysis, we found that the expression levels of *FvF3H48, FvF3H120* and *FvF3H74* were higher in the stages of germination, growth, flowering and fruit setting. The expression levels of *FvF3H42* and *FvF3H16* were higher in seeds. The expression levels of *FvF3H16* and *FvF3H11* were higher in the ovary wall of stage 1, stage 2, stage 3 and stage 5. *FvF3H15* and *FvF3H48* were highly expressed in the pericardium, anther, receptacle and anther. Real-time fluorescence quantitative PCR showed the expression changes in *F3H* in the fruit coloring process. The results indicate that the expression levels of most members were higher during the S3 stage, such as *FvF3H7, FvF3H16, FvF3H32, FvF3H82, FvF3H89, FvF3H92* and *FvF3H112. FvF3H63* and *FvF3H104* exhibited particularly high expression levels during the S1 stage, with some genes also showing elevated expression during the S4 stage, including *FvF3H13, FvF3H27, FvF3H66* and *FvF3H103. FvF3H58, FvF3H69, FvF3H79* and *FvF3H80* showed higher expression levels during the S2 stage. These findings lay the groundwork for elucidating the biological functions of the strawberry *F3H* gene family and the selection of related genes.

## 1. Introduction

Color is an important part of the appearance quality of strawberry berries, which not only determines the market value of fresh strawberries, but also affects their processing purposes and the quality of processed products and has great economic value [1]. Extensive studies have shown that fruit color is caused by plant pigments, including lycopene cords, anthocyanins and carotenoid cords [2]. The color span of strawberries is large, from completely white to deep red, and the formation of fruit color is mainly because of the different content of anthocyanins. The proportion of anthocyanins in strawberries and the different accumulation levels make strawberries show red, white or pink, yellow and so on [3].

Anthocyanins are one of the flavonoid compounds, a class of water-soluble pigments widely present in plant vacuoles, mainly in the form of stable polyglycosides, collectively known as anthocyanins. Anthocyanin biosynthesis is a branch of the plant flavonoid synthesis pathway, which is also catalyzed by phenylalanine lyase (PAL), cinnamate hydroxylase (C4H) and coumaric acid CoA ligase (4CL) to form 4-coumaryl CoA as in other plants [4]. Subsequently, 4-coumaryl CoA was catalyzed by chalcone synthetase (CHS) to produce yellow chalcone, which was catalyzed by chalcone isomerase (CHI) and flavanone 3-hydroxylase (F3H) to form dihydroflavonol. Dihydroflavonol synthesized in flavonoid 3′-hydroxylase (F3′H) and flavonoid 3′5′-hydroxylase (F3′5′H) catalyzed the formation of the precursors of anthocyanin synthesis. Dihydroquercetin and dihydromyricetin were catalyzed to form colorless anthocyanin via the action of dihydroflavonol-4-dehydrogenase (DFR). The catalysis of colorless anthocyanin dioxygenase/anthocyanin synthetase (LDOX/ANS) forms colored anthocyanin. Finally, anthocyanins form glycosidic bonds with one or more glucose, rhamnose, galactose, xylose and arabinose under the action of glucosyltransferase and are finally transformed into stable anthocyanins [5].

This process is regulated by several structural genes (including *PAL*, *C4H*, *4CL*, *CHS*, *CHI*, *F3H*, *F3′H*, *F3′5′H*, *DFR, ANS*, etc.) [6]. The *F3H* gene is involved in anthocyanin accumulation and expression to regulate flower color. In *Chrysanthemum × morifolium* Ramat, the *F3H* gene is a key enzyme responsible for regulating the metabolism of flavonoids and anthocyanins [6]. Inactivation of *F3H* gene mutation can block the anthocyanin synthesis pathway in petunias, thus producing white flowers [7]. Antisense RNA technology was used to inhibit the *F3H* gene of *Dianthus caryophyllus* L. to make the flower color from orange–red to pale or even white, and the presence of anthocyanins in white plants was not detected [8]. The expression of *F3H* gene in *Nelumbo nucifera* with red, purple and blue petals is high, and the expression of *F3H* increases when petals change from white to pink [7]. The expression of *F3H* gene and its expression intensity are also the key factors for anthocyanin synthesis in fruit. The expression of *RrF3H* increased with the deepening of fruit color and reached the highest value when the fruit was about 75% colored [9]. The expression of *RaF3H* in *R. albrum* L. decreased gradually, and the expression of *F3H* in *R. albrum* L. was higher than that in *R. albrum* L. [9]. Through RT-PCR analysis of three different colored mangoes *(Mangifera indica)*, it was found that the expression of *F3H* in the peel of red mango was the highest, followed by yellow Jinhuang variety, and the lowest was green Guiqi variety [10]. Compared with *Dimocarpus longan* Lour, genes such as *F3H* in red fruit longan are significantly up-regulated, which results in anthocyanin accumulation in the skin, presenting a strong red color [7]. When the *F3H* gene was introduced into sand pear fruit, the fruit color turned red, indicating that this gene regulates the development and formation of color in pear fruit [11].

Although the function of the *F3H* gene in regulating color formation has been studied in many plants, the identification of *F3H* gene family members and their expression characteristics in different coloration stages of forest strawberry have not been reported. In this paper, the *F3H* gene family members of strawberry were identified using the bioinformatics method. The structure, collinear relationship, phylogeny, *cis*-acting elements and tissue expression patterns of *F3H* gene family members were analyzed. This study has laid the foundation for further investigation into the biological functions and molecular mechanisms of *F3H* gene members in strawberries.

## 2. Results

### 2.1. Identification and Physicochemical Properties of F3H Gene Family in Strawberry

Using the amino acid sequence of *F3H* gene in *Arabidopsis thaliana* as the query sequence, a total of 126 genes were retrieved using TBtools blast and NCBI protein blast, and the 126 *F3H* genes were named *FvF3H1–FvF3H126* according to their positions on chromosomes. The shortest amino acid length is 151aa (*FvF3H28*), the longest is 608aa (*FvF3H20*) and the molecular weight is between 17122.77 Da and 68294.08 Da. The isoelectric point ranges from 4.77 (*FvF3H84*) to 9.65 (*FvF3H28*). Except for basic proteins with theoretical isoelectric points greater than seven, such as *FvF3H28*, *FvF3H85*, *FvF3H100*, *FvF3H47* and *FvF3H77*, all proteins are acidic. According to the analysis of physical and chemical properties of proteins, it is predicted that the family members may play different functions (Appendix A).

### 2.2. Phylogenetic Tree, Secondary Structure and Subcellular Localization of Strawberry F3H Family

The amino acid sequences of 126 *F3H* genes of strawberry were used for phylogenetic analysis and were divided into six subfamilies according to evolutionary relationship (Figure 1). Group 6 subfamily had the most genes, including 54 members, while group 3 subfamily had the least genes, including only 6 members. The secondary structure prediction (Appendix A) showed that all genes had no β-corner, mainly α-helix, random curling and extension chain. The most is random crimp (15.95−62.11%), followed by α-helix (15.95−48.10%), and the least is extended chain (9.94−26.51%). The subcellular localization of *F3H* gene family indicated that the proteins encoded by *F3H* gene family were mainly located in the cytoplasm, chloroplast, nucleus, mitochondria and cytoskeleton.

### 2.3. Analysis of Gene Structure, Motif, Domain and Cis-Acting Elements

According to gene structure analysis (Figure 2), 31 *FvF3H* genes did not contain upstream and downstream sequences, and the number of exons ranged from one to eight. *FvF3H117* contains only one exon, while *FvF3H20* and *FvF3H21* contain eleven exons. Most genes contain two to four exons. On the MEME website, the conserved motifs of the *F3H* gene family proteins are predicted, which include a total of 15 motifs. The N-terminus of most sequences is motif12, and the C-terminus is motif9. *FvF3H44* contains two instances of motif5. *Cis*-acting elements were analyzed for the first 2000 bp of the strawberry *F3H* gene promoter. *F3H* gene mainly contained light, hormone, abiotic stress and meristem response elements. Hormone response elements contained auxin, gibberellin, abscisic acid and salicylic acid response elements, and abiotic stress response elements contained low temperature, drought, defense and stress and wound response elements.

### 2.4. Chromosome Localization and Collinearity Analysis

Chromosome localization analysis was performed using the MG2C website, and 126 members were distributed on seven chromosomes. In chromosome 1, there are 18 genes; chromosome 2 has 27 genes; chromosomes 3 and 5 each have 17 genes; 15 genes are located on chromosome 4; 13 genes are on chromosome 6; and only 9 genes are on chromosome 7. Chromosome 2 has the highest gene distribution, accounting for 21.43% of the total genes, while chromosome 7 has the lowest gene distribution, each accounting for 7.14% of the total genes (Figure 3).

To further understand the evolutionary relationship of gene families, the MCScanX tool of TBtools was used to conduct collinearity analysis within and between species. A total of six collinearity relationships were found in *F3H* gene family species, which were located on chromosomes chr1, chr2, chr4, chr5, chr6 and chr7, respectively. They are *FvF3H83/FvF3H122, FvF3H78/FvF3H18, FvF3H74/FvF3H37, FvF3H70/FvF3H34, FvF3H118/FvF3H123* and *FvF3H122/FvF3H126.* These results suggest that some *F3H* genes may be produced through gene replication, and these genes may have similar functions (Figure 4).

The collinear relationship maps of strawberry with *Arabidopsis thaliana*, grape, apple and rice were drawn, and the homologous genes with *Arabidopsis thaliana*, grape, apple and rice were 27, 45, 66 and 8 pairs, respectively, indicating that strawberry and dicotyledonous plants had more homologous genes than monocotyledonous plants (Figure 4).

### 2.5. Codon Bias

The constituent indexes of codon include “CAI” as codon adaptation index; “CBI” means codon bias index; “FOP” is the frequency of the occurrence of the optimal codon; “Nc” is the number of valid codons; “GC” is the gene count (G+C); “GC3s” is the number of the third codon (G+C). The frequency of relative synonymous codons in strawberry genome was analyzed, and it was found that the RSCU of 32 codons was ≥1. Namely GGC, GGU, GAG, GAU, GCA, GCU, GUG, GUU, AGG, AGA, AGC, AAG, AAC, ACA, ACC, ACU, AUG, AUC, AUU, CAA, CAU, CCA, CCU, CUC, CUU, UGA, UGC, UAC, UCA, UCU, UUG and UUC. Among them, there are nine instances where the third codon is C, ten instances where it is U, six instances where it is G and the remaining seven instances where it is A. This indicates that the third codon of the amino acid of the strawberry F3H protein is more inclined to C or U (Figure 5). The average values of CAI, CBI, Fop and Nc in strawberry *F3H* family members were 0.21, −0.04, 0.40 and 54.92, respectively. The GC content of *FvF3H* family members ranged from 41.9% to 59.70%, and the GC3s content ranged from 33.40% to 76.60%, with average values of GC and GC3s being 45.86% and 46.08%, respectively. A total of 13 genes were found to have Nc values less than 50; they are *FvF3H2*, *FvF3H3*, *FvF3H8*, *FvF3H15*, *FvF3H23*, *FvF3H31*, *FvF3H32*, *FvF3H57*, *Fv3H77*, *FvF3H80*, *FvF3H88*, *FvF3H103* and *FvF3H119*, respectively (Appendix A). It shows that the codon preference of these 13 genes is strong. The correlation graph shows that T3s is positively correlated with A3s and negatively correlated with C3s, G3s, CAI, CBI, Fop, Nc, GC and GC3s. C3s is negatively correlated with T3s and A3s. CAI, CBI, Fop, GC and GC3s are positively correlated with Nc and negatively correlated with T3s and A3s. It was positively correlated with C3s, GC and GC3s (Figure 6).

### 2.6. Tissue-Specific Expression Analysis and Protein Interaction Prediction

The expression patterns of *FvF3H* gene family members were analyzed during the whole development period of the plant, including seeds (ovary wall, embryo, endosperm and seed coat tissue), young leaves, seedlings, different tissues in flowers (perianth, carpellary, inner pellary, fleshy tissue below achene) and pollen (and pollen microspore) (Figure 7). It was found that genes in the same subfamily had similar expression levels. The expression levels of *FvF3H9* and *FvF3H114* were higher in leaves, but lower in other tissues. The expression levels of *FvF3H48, FvFH120* and *FvF3H74* were higher in flower stages 1–4. The expression levels of *FvF3H42* and *FvF3H16* were higher in the second stage. The expression levels of *FvF3H16* and *FvF3H11* were higher in the ovary wall of stage 1, stage 2, stage 3 and stage 5. *FvF3H15* and *FvF3H48* were highly expressed in the carpellum, anther and receptacle.

The interactions between 126 *FvF3H* proteins were predicted using STRING online software (Figure 8). The results showed that there might be interaction among 52 *FvF3H* proteins. Most *FvF3H* proteins form a complex network structure, such as *FvF3H18*, *FvF3H45*, *FvF3H57*, *FvF3H123*, etc. *FvF3H17*, *FvF3H104*, *FvF3H112*, *FvF3H115* and *FvF3H124* interact with XP_004299308.1 (flavonoid 3′-monooxygenase-like). *FvF3H12*, *FvF3H82*, *FvF3H39* and *FvF3H126* interact with XP_004307734.1 (chalcone-flavonoid isomerase 3). *FvF3H17*, *FvF3H39*, *FvF3H104*, *FvF3H115* and *FvF3H126* interact with XP_004309662.1 (anthocyanin reductase). XP_00429930801 (flavonoid 3′-monooxygenase-like), XP_004307734.1 (chalcone-flavonoid isomerase 3) and XP_004309662-1 (anthocyanin reductase) are involved in the biosynthesis and molecular regulation of anthocyanin in plants. The above genes are closely related to anthocyanin synthesis.

### 2.7. Determination of Anthocyanin Content and Expression Analysis of FvF3H Gene Family in Strawberry at Different Coloring Stages

The different stages of strawberry coloring, namely the green fruit stage, 20% coloring stage, 50% coloring stage and complete coloring stage, can be observed from S1 to S4. Anthocyanin content gradually increases as the fruit coloring progresses (Figure 9). The qRT-PCR analysis (Figure 10) revealed the expression of the *FvF3H* gene across all stages, suggesting the potential involvement of the *F3H* gene family in various stages of strawberry anthocyanin accumulation or biosynthesis. However, irregular variations were observed in the expression levels during different growth stages. Most genes exhibited their highest expression levels during the S3 period. For instance, the expression level of the *FvF3H16* gene during the S3 period was 70 times higher than that during the S1 period, while the *FvF3H112* gene showed an expression level in the S3 period that was 118 times higher than in the S1 period. Similarly, the *FvF3H82* gene had an expression level during the S3 period that was 15 times higher than during the S1 period, and the expression of the *FvF3H89* gene during the S3 period was 16 times that of the S1 period. However, there were some differences observed, such as the expression of *FvF3H19* being 24 times higher in S1 compared to S2. Moreover, significant differences were noted in the expression levels of *FvF3H103* and *FvF3H13* between S4 and S2, with the expression level of *FvF3H13* in S4 being notably higher than that in S2. There is a significant difference between the S1 and S4 phases of *FvF3H104*, with the expression level in S1 being 87 times higher than that in the S4 phase.

## 3. Discussion

The cDNA of *F3H* gene was originally cloned from *Antirrhinum majus* and has been cloned in many plants, such as apple and *Medicago sativa* [12]. Since the substrate of *F3H* is naringenin, *F3H* regulates the synthesis of flavonoid and anthocyanin glycoside products and is the central site of the entire flavonoid metabolic pathway [13]. In this study, the *F3H* genome of strawberry was analyzed using the bioinformatics method, and 126 members of the *F3H* family were identified, which is a large gene family compared with the family members in fruit *Chrysanthemum × morifolium* Ramat and *Triticum aestivum* L. [14,15]. Chromosome localization showed that *F3H* gene was unevenly distributed on seven chromosomes of strawberry (Figure 3). Some genes form gene clusters on chromosomes, which are speculated to be formed by tandem repeats, and it is speculated that tandem repeats may be the main reason for the expansion of strawberry *F3H* family. Subcellular localization prediction of coding proteins (Appendix A) found that most of the coding proteins of *FvF3H* family members were located in the chloroplasts, cytoplasm, nucleus and cytoskeleton, and a few were located in the endoplasmic reticulum, mitochondria, Golgi apparatus and vacuoles. These results are consistent with subcellular localization results of barley, *Hordeum vulgare* var. *coeleste* Linnaeus, *Sorghum bicolor* (L.) Moench, *Triticum aestivum* L., *Zea mays* L., *Allium cepa* L., *Anthurium andraeanum* Linden and *Lilium candidum* L. [6]. The gene pairs with collinear relationships in the *F3H* gene family include *FvF3H83/FvF3H122*, *FvF3H70/FvF3H34*, *FvF3H78/FvF3H18*, *FvF3H74/FvF3H37*, *FvF3H118/FvF3H123* and *FvF3H122/FvF3H126* (Figure 4). *FvF3H122* has two tandem repeats, and *FvF3H78* and *FvF3H18*, *FvF3H118* and *FvF3H123* are in subgroup 2, *FvF3H70* and *FvF3H34* are in subgroup 6 and *FvF3H122* and *FvF3H126* are in subgroup 5. It is suggested that the homology is high, the gene structure and conserved motifs are very similar and these genes may have similar functions. In addition, we explored the collinearity of strawberry *F3H* gene with *Arabidopsis*, apple, grape and rice (Figure 4). The results showed that there were more homologous gene pairs between strawberry and dicotyledon than between strawberry and monocotyledon. We believe that strawberry and dicotyledonous plants have a closer phylogenetic relationship.

The promoter of a gene may determine the function of a gene. In this study, *cis*-acting elements were analyzed on the first 2000 bp sequence of *FvF3H* gene, and it was found that there were more elements responding to light, hormone and abiotic stress, indicating that *F3H* gene was also involved in the response to abiotic stress (Figure 2). An et al. [16] found that with the increase in light intensity and development stage, the anthocyanin content in blueberry leaves showed a trend of first increasing, then decreasing and then increasing. Appropriate light intensity can significantly promote anthocyanin synthesis. Under natural light conditions during the day, low temperature at night reduces the activity of UFGT (UDP-glucose: flavonoid 3-O-glucosyltransferase) by affecting the metabolism of ascorbic acid, thus reducing the concentration of anthocyanins in Fuji apple peel [17]. The expressions of *F3H* and *FLS* genes were significantly up-regulated in drought-resistant and drought-sensitive potato strains, and the expression patterns of genes related to stress response were also different [18]. Enrique et al. [19] revealed the key role of *F3H* in the metabolism of flavonoids in blackberry, thereby improving the adaptability of blackberry to biological stress.

The analysis of gene tissue expression showed that the expression of *F3H* gene family members in different tissues was different (Figure 7). The expression levels of *FvF3H9* and *FvF3H114* are high in leaves and low in other tissues, and the color of leaves is darker in the young stage, which may be due to the induction of anthocyanin synthesis pathway-related genes to express and accumulate more anthocyanins, thereby protecting young leaves, such as reducing the photo-inhibition phenomenon [20]. The expression levels of *FvF3H48, FvFH120* and *FvF3H74* were higher in flower stages 1–4. In *Hydrangea macrophylla*, the expression level of *HmF3H* gene in flowers is significantly higher than that in roots, stems, leaves and other tissues and organs, and there are also significant differences in *HmF3H* gene expression in different flower varieties and different development stages. The expression level of *HmF3H* gene in dark flowers is significantly higher than that in light flowers, which may be related to the accumulation of more anthocyanidins in dark flowers [21]. The study on the expression of *CnF3H* in different developmental stages and different parts of flower organs showed that the expression level of *CnF3H* gene was high in the young bud stage, the first bud stage and the color stage of flower development, and the expression level was gradually decreased with the opening of flowers [22]. Tissue-specific expression analysis of *SoF3H* gene in *Syringa oblata* Lindl. showed that the expression level of *SoF3H* gene was the highest in flowers and the highest in flower bud stage [23]. The expression levels of *FvF3H16* and *FvF3H11* were higher in the ovary wall of stage 1, stage 2, stage 3 and stage 5. *FvF3H15* and *FvF3H48* were highly expressed in carpellum, anther and receptacle. The tissue-specific expression analysis of *Bletilla striata* showed that the expression of *Bletilla striata* was the highest in flowers, the second in leaves and the least in the stems, tender capsules and tuberous bulbous bulbs, which was consistent with the above results [24].

Understanding the biological function of unknown or known proteins can reasonably predict the cellular function of proteins. Protein interaction prediction results showed that some genes interact with genes related to flavonoid synthesis, such as flavonoid 3′-monooxygenase-like, chalcone-flavonoid isomerase 3 and anthocyanin reductase (Figure 8). It was found that *HvCHI* was highly expressed in the flower of hostan and was positively correlated with anthocyanin content, and the overexpression of *HvCHI* in transgenic tobacco promoted anthocyanin accumulation [25]. Studies in *Morus alba* L. found that the expression levels of *MmCHI1* and *MmCHI2* were positively correlated with anthocyanin content during fruit ripening [26]. Kumari et al. [27] found that 3′-monooxygenase-like monooxygenase was mainly involved in kaempferol degradation.

Flavanone 3-hydroxylase (*F3H*) connects key nodes in the downstream branches of the anthocyanin synthesis pathway. Therefore, the *F3H* gene plays an important role in the color formation process of different plant tissues [28]. In this study, real-time fluorescence quantitative PCR was used to analyze the relative expression of *F3H* in four stages of fruit coloring, and the results showed that the expression of *F3H* could be detected during the whole fruit development process (Figure 10). From the green fruit stage to the 50% color change stage, the expression of *FvF3H* gradually increases. Although the anthocyanin content continues to increase afterward, the gene expression level decreases. This suggests that during fruit development, the expression of *FvF3H* increases with the maturation and deepening of fruit coloration, providing a large amount of precursor material for the synthesis of auxiliary pigments such as anthocyanins in the fruit. In addition, since flavanone 3-hydroxylase plays a key role in the early stage of anthocyanin synthesis, the increase in anthocyanin content in the later stage may be caused by other structural genes, such as dihydroflavonol-4-reductase (DFR), anthocyanin synthetase/achromoanthocyanin dioxygenase (ANS/LDOX) and UFGT, MYB transcription factors associated with anthocyanin synthesis regulate structural gene expression, or some environmental factors. Studies on anthocyanin synthesis in pear and apple fruit coloring have shown that the accumulation of anthocyanins in late fruit ripening is mainly due to the key transcription factor MYB regulating the expression of structural gene UFGT, thus affecting anthocyanin production [29,30]. Building upon this knowledge, our investigation in strawberries aimed to quantify the expression levels of key genes involved in anthocyanin synthesis. qRT-PCR analysis revealed that, in strawberries, the expression levels of most genes were notably high in the S3 stage. Noteworthy genes such as *FvF3H7*, *FvF3H32*, *FvF3H82*, *FvF3H83*, *FvF3H89* and *FvF3H112* were identified, indicating their significant role in the fruit coloring stage. The expression of *F3H* gene in mango varieties with red skin color was higher than that in Jinhuang varieties with yellow skin color at ripening stage than that in Guiqi varieties with green skin color at ripening stage, indicating that *F3H* gene plays an irreplaceable role in anthocyanin anabolic pathway [31]. Yang et al. [32]. analyzed the expression of *AcF3H* gene in the fruits of ‘Hongyang’ kiwifruit at different developmental stages. The results showed that the expression of *AcF3H* gene was high before fruit color transformation, but decreased at the beginning of fruit color transformation and then maintained at a high level with the deepening of fruit color. However, some genes were still highly expressed in S1, S2 and S4 periods. For example, the expression level of *FvF3H6* was higher in S2 period, and the expression level of *FvF3H13* was highest in S4 period. Possibly because different genes play a role at different times, the detailed function of the *FvF3H* gene has not been verified, and it remains to be studied in plant pigment function.

Anthocyanin is a kind of water-soluble natural pigment widely existing in plants in nature. There are more than 250 kinds of naturally occurring anthocyanins, which exist in 27 families and 73 genera of plants. Twenty anthocyanins have been identified, six of which are common in plants, namely, geranium pigments (Pg), centaurea pigments (Cy), delphinium pigments (Dp), paeoniflorin (Pn), morning glory pigments (Pt) and malvins (Mv). In this study, only the total amount of anthocyanin in strawberry fruits was measured, and the expression of *F3H* gene family members in strawberry at different coloring stages was analyzed. According to the relationship between the relative expression level in different periods, the change in fruit color and the total content of anthocyanin, the key candidate *F3H* members involved in anthocyanin synthesis were preliminarily identified. The expression level of each gene family member must have a certain relationship with the content of a single component of anthocyanin. In subsequent studies, it is more important to analyze the correlation between the expression level of *F3H* members in strawberry and the synthesis of single anthocyanin components, so as to determine the regulatory relationship between key genes and key anthocyanin components.

## 4. Materials and Methods

### 4.1. Plant Materials

“Monterey” strawberry fruit was selected as the research material, and the green fruit, 20% coloring stage, 50% coloring stage and fully coloring stage were collected. A total of 15 fruit samples were collected at different coloring stages, and every fifth fruit was a duplicate, accurately weighed, quickly frozen with liquid nitrogen and stored at −80°C for subsequent experiments.

### 4.2. Identification of F3H Gene Family in Strawberry

*Arabidopsis thaliana* database (https://www.arabidopsis.org/, accessed on 10 accessed July 2023) was used to obtain *F3H* gene sequences of proteins. Phytozomev13 (https://phytozome.jgi.doe.gov/pz/portal.html, accessed on 12 July 2023) website strawberry genome and annotation file was used, TBtools software (version 1.108) was used to extract all protein sequences of strawberry [33] and bidirectional blast comparison was conducted with Arabidopsis family protein sequences to preliminarily obtain *F3H* family members of strawberry [34]. Then, the preliminary screening of protein sequence was performed using the NCBI protein blast plate for their second blast, and the NCBI-CDD website (https://www.ncbi.nlm.nih.gov/cdd/, accessed on 15 July 2023) was used to analyze protein conservative structure domain, removing sequence fragments with incomplete or missing domains. Then, combined with the 2OG-FeⅡ_Oxy (pfam03171) functional domain, 126 strawberry *F3H* genes were retrieved, and their gene length, coding sequence length (CDS) and amino acid sequence were downloaded.

The ExPASy tool (https://web.expasy.org/protparam/, accessed on 5 August 2023) [35] attained the physical and chemical quality, such as hydrophilic high average (GRAVY), isoelectric point (PI), molecular weight (MW), instability index (II) and fat index (AI).

### 4.3. Evolutionary Tree Construction, Secondary Structure and Subcellular Localization

ClustalX 1.83 software was used for multiple sequence comparison, MEGA 7.0 software was used to draw the evolutionary tree and the adjacent method (NJ) was adopted for construction. The bootstrap value was 1000. The EVOLVIEW website (https://evolgenius.info//evolview-v2/#login, accessed on 7 August 2023) was used to beautify [36]. Secondary structure prediction was made using the website NPS@:SOPMA (https://npsa-prabi.ibcp.fr/cgi-bin/npsa_automat.pl?page=npsa_sopma.html, accessed on 10 August 2023). The WoLF PSORT (https://wolfpsort.hgc.jp/, accessed on 10 August 2023) website was used for subcellular localization analysis [37].

### 4.4. Analysis of Gene Structure, Motif and Cis-Acting Elements

Gene structure prediction was constructed using TBtools software (Version 1.108). The conserved motifs of proteins were constructed using MEME (http://meme-suite.org/tools/meme, accessed on 15 August 2023), the number of motifs was set to 10 and the remaining parameters were all default values [38]. The 2000 bp upstream sequence of the *FvF3H* gene was obtained using TBtools software with online software PlantCARE (http://bioinformatics.psb.ugent.be/webtools/plantcare/html/, accessed on 18 August 2023) and plotted at TBtools (Version 1.108).

### 4.5. Chromosome Localization and Collinearity Analysis

The chromosomes of *F3H* family members of strawberry were mapped using MG2C (http://mg2c.iask.in/mg2c_v2.0/, accessed on 19 August 2023). To analyze *F3H* gene collinearity relationship, for a total of linear analysis of *Arabidopsis thaliana*, apples, grapes and the rice genome, the annotation files were downloaded from phytozomev13 (https://phytozome.jgi.doe.gov/pz/portal.html, accessed on 19 August 2023). The gene pairs of the strawberry *F3H* gene pairs were identified utilizing the collinearity tool within TBtools (Version 1.108) and subsequently visualized [39].

### 4.6. Codon Bias

CodonW1.4.2 (http://codonw.sourceforge.net) online software analysis *F3H* codon was used for the characteristics of gene sequences of CDS, which include relative synonymous codon usage (RSCU), effective codon (ENC), codon bias index (CBI), codon adaptation index (CAI), optimal codon usage frequency (Fop), T3s, C3s, A3s, G3s and more. T3s, C3s, A3s, G3s, CAI, CBI, Nc, Fop, GC, GC3s, L_sym, L_aa, GRAVY and Aromo parameter correlation analysis was also performed.

### 4.7. Tissue-Specific Expression Analysis and Protein Interaction Prediction

The expression levels of *F3H* gene in different tissues of strawberry were retrieved in BAR database (https://bar.utoronto.ca/#, accessed on 26 August 2023), including pollen, anther, style, fleshy tissue, flower, receptor, carpellum, leaf, etc. Log10 transformation was performed on the selected data, and plots were performed in TBtools (Version 1.108). The protein interaction network was predicted using STRING version 11(https://string-db.org/, accessed on 25 August 2023) [40].

### 4.8. Determination of Anthocyanin Content in Strawberry Peel at Different Developmental Stages

Fifteen strawberry fruits were selected from each development stage, and for each 3 fruits, there was 1 biological replicate. Each repeated fruit was homogenized, accurately weighed to 1.0 g and ground with liquid nitrogen for the determination of anthocyanin. The specific method was referred to and combined with the pH difference method of Dussi et al. (1995), Tao et al. (2018) and Jeong et al. (2004) and pre-delivered in the form of mg anthocyanin-3-galactoside per 100 g fresh tissue. The homogenate was placed into a 10 mL centrifuge tube, the mortar was rinsed with 1% hydrochloric methanol solution, and it was transferred to the test tube, rinsing the mortar with 1% HCl-methanol solution. The volume was fixed to the scale and then mixed. Extraction was carried out at 4 °C for 20 min in the dark, during which the extraction was shaken 5−10 times for 10−30 s each time. Samples were then filtered through 0.2 µm PES filters (Krackeler Scientific, Inc., Albany, NY, USA) and analyzed using TU-1900 double beam UV–visible spectrophotometer (Beijing Purkinje General Instrument Co. LTD, Beijing, China). The solution was zeroed with 1% HCl-methanol solution as blank reference, and the absorbance of the solution was determined with filtrate at 600 nm and 530 nm, respectively. Anthocyanin content (U) was expressed by the difference in absorbance values at wavelengths 530 nm and 600 nm per gram of fresh weight peel tissue, i.e., U = (OD_530_ − OD_600_)/gFW.

### 4.9. qRT-PCR Analysis

The primers (Appendix A) were synthesized by Shenggong Bioengineering Co., LTD. (Shanghai, China) RNA was extracted from strawberry fruit and reverse-transcribed into single-strand cDNA as template. The quantitative reaction system consisted of 20 μL: 1 μL cDNA, 1 µL of each of the upstream and downstream primers (10 µmol/L), 10 μL SYBR enzyme and 7 μL ddH_2_O. The cycle parameters were 30 s at 95 °C, 5 s at 95 °C for 40 cycles and 34 s at 60 °C. The melting curve analysis was performed after the PCR cycle, and the procedures included 95 °C for 15 s, 60 °C for 60 s and 95 °C for 15 s. Three biological replicates and technical replicates were set up in this experiment. The melting curve and fluorescence value change curve were analyzed after the reaction procedure. The relative expression of genes was calculated using 2^−ΔΔ*Ct*^ [36].

### 4.10. Statistical Analysis of Data

Statistical data were analyzed using Excel software (Version 2019), calculated and sorted out. Three repeated qRT-PCR quantitative data were analyzed via one-way analysis of variance using SPSS 22.0. A *p* < 0.05 was significant difference.

## 5. Conclusions

In this study, 126 *F3H* genes of strawberry were identified, which were distributed unevenly in seven chromosomes and could be divided into six subfamilies according to evolutionary relationship. Protein interaction prediction results showed that some of the genes were related to flavonoid 3′-monooxygenase, chalcone-flavonoid isomerase 3 and anthocyanin reductase, which could jointly regulate anthocyanins synthesis. The results of qRT-PCR showed that the expression levels of *FvF3H7*, *FvF3H16*, *FvF3H112*, *FvF3H97* and *FvF3H82* were higher in the rapid coloring stage, and the expression of *FvF3H58* was higher in the early stage of color change. These genes can be used as candidate genes for further functional studies. This study will contribute to a better understanding of the *F3H* gene family’s role in the color changes in strawberries, laying the foundation for further exploration of the biological functions and molecular mechanisms of *F3H* gene members in strawberries.

## Figures and Tables

**Figure 1 ijms-24-16807-f001:**
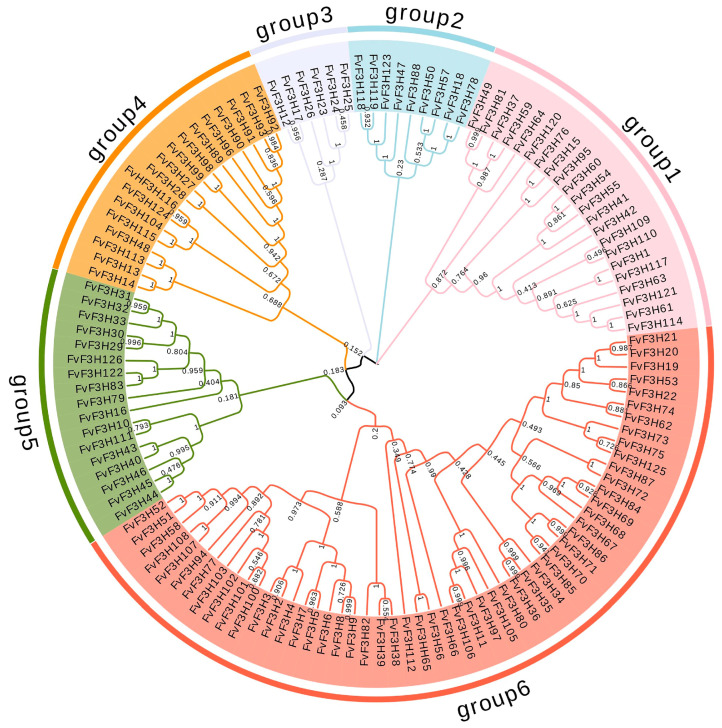
Phylogenetic analysis of the strawberry *FvF3H* gene family. Phylogenetic trees were constructed using the F3H protein sequences. NJ method was adopted, and the bootstrap value was set to be equal to 1000.

**Figure 2 ijms-24-16807-f002:**
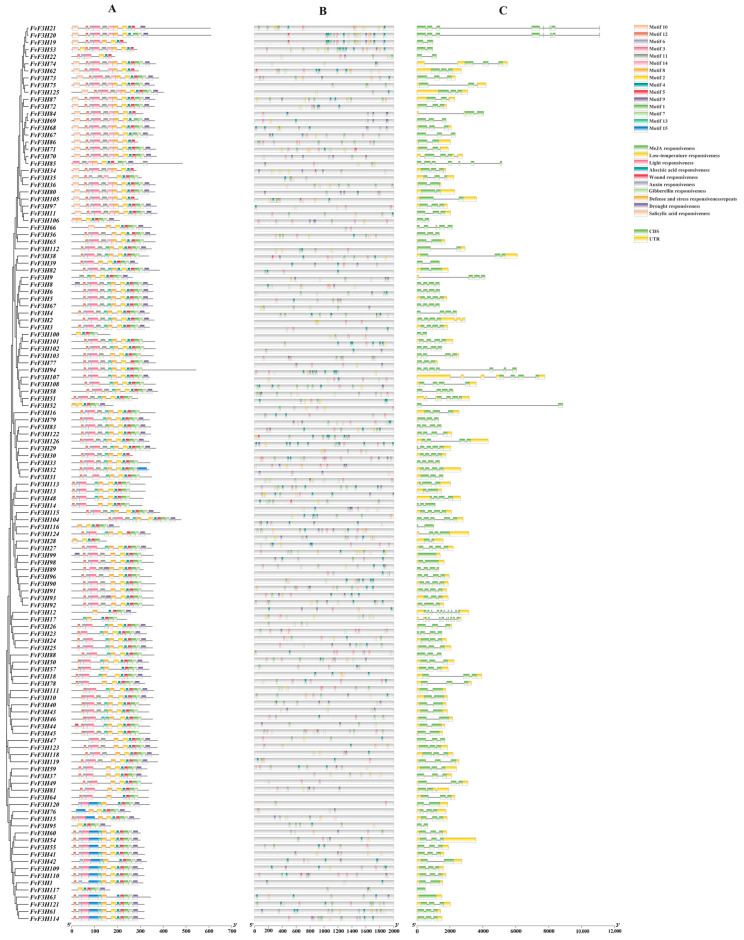
Motif, *cis*-regulatory element analysis and gene structure analysis of *FvF3H* gene. (**A**) Analysis of conserved motif of *F3H* gene in strawberry. (**B**) *Cis*-regulatory element analysis of the *FvF3H* genes. (**C**) The exon–intron structure of *FvF3H* genes.

**Figure 3 ijms-24-16807-f003:**
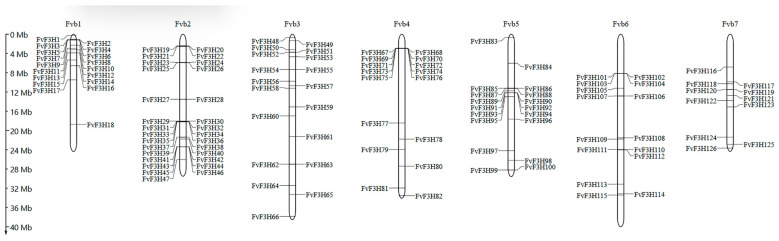
Chromosome distribution of the *F3H* gene family in strawberry. The left scale indicates the chromosome length (Mb), with *F3H* gene markers on the right side of each chromosome.

**Figure 4 ijms-24-16807-f004:**
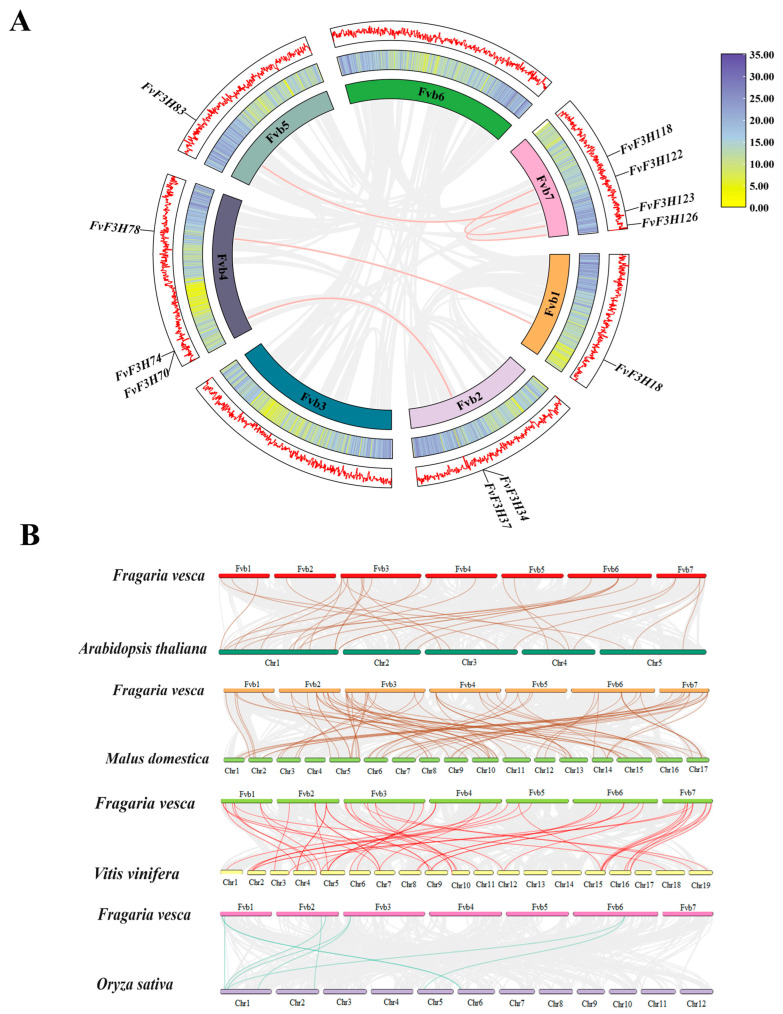
Collinearity analysis of *F3H* gene families. (**A**) Collinearity analysis of *FvF3H*. The gray lines represent all collinear blocks in the strawberry genome, and the pink lines represent gene pairs between the *FvF3H* genes. (**B**) Collinearity analysis of *F3H* gene in strawberry and four representative plants. The gray lines in the background show collinearity between the strawberry and *Arabidopsis thaliana*, grape, apple and rice genomes. The light brown lines show collinearity between the *FvF3H* gene and *Arabidopsis thaliana*, the brown lines show collinearity between the *FvF3H* gene and apple and the red lines show collinearity between the *FvF3H* gene and grape. The green lines represent collinear gene pairs between the *FvF3H* gene and rice.

**Figure 5 ijms-24-16807-f005:**
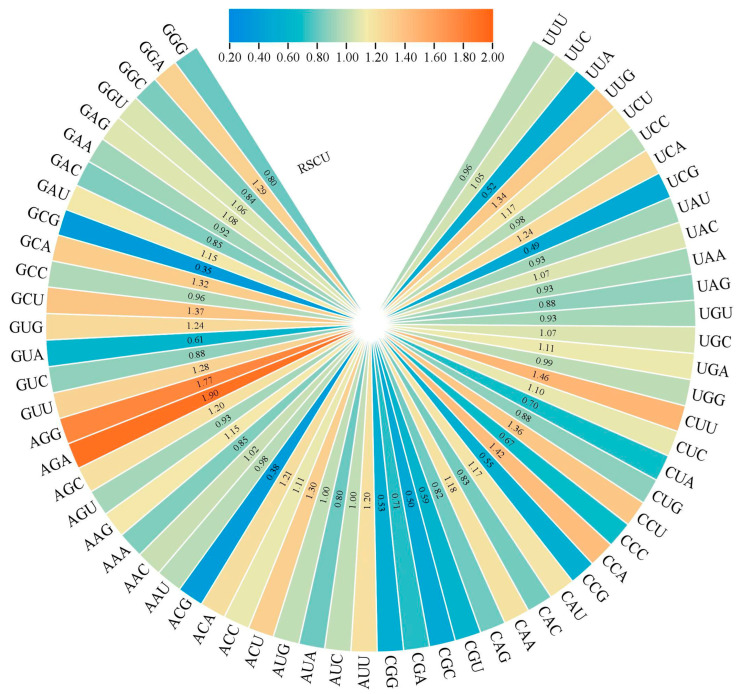
Relative synonymous codon usage (RSCU) analysis of *F3H* gene codon in strawberry.

**Figure 6 ijms-24-16807-f006:**
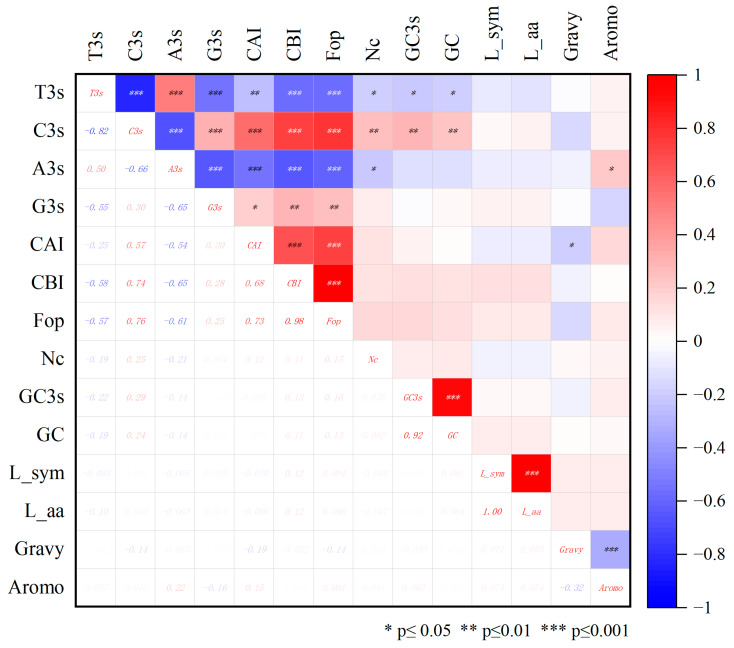
Codon correlation analysis of strawberry *F3H* gene. Blue indicates positive correlation, red indicates negative correlation and white indicates no correlation. The darker the color, the stronger the correlation, and vice versa.

**Figure 7 ijms-24-16807-f007:**
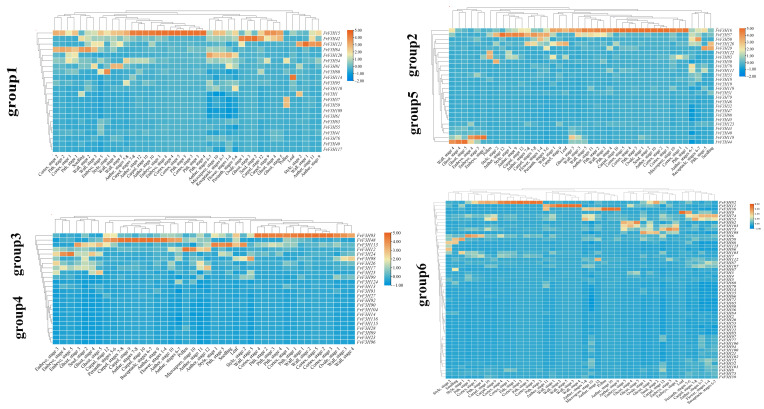
Expression of *F3H* gene in different tissues of strawberry. The numbers behind different tissues indicate developmental stages. Red or blue shading represent the up-regulated or down-regulated expression level, respectively. The scale denotes the relative expression level.

**Figure 8 ijms-24-16807-f008:**
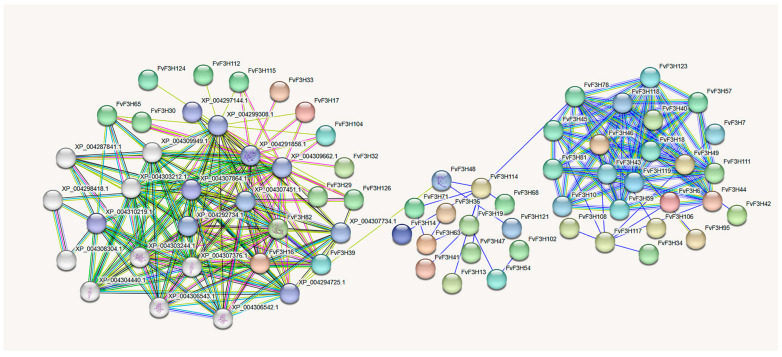
Analysis of protein interaction of *F3H* gene family in strawberry. Nodes indicate proteins. Empty nodes indicate the protein of unknown 3D structures, and filled nodes indicate that some 3D structures are known or predicted. The connection between nodes indicates the interaction between proteins, and different colors correspond to different types of interactions.

**Figure 9 ijms-24-16807-f009:**
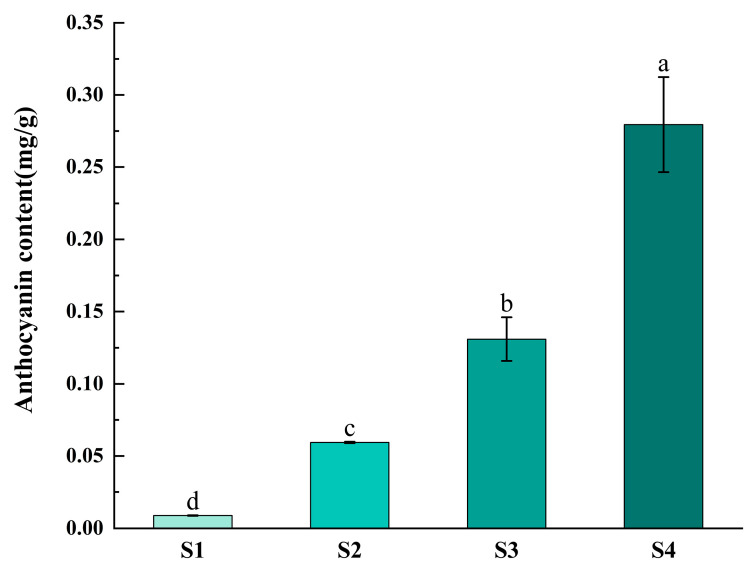
Expression analysis of strawberry anthocyanins in four periods. S1 represents the green fruit stage, S2 represents the 20% coloration stage, S3 represents the 50% coloration stage and S4 represents the complete coloration stage. Different letters denote significant differences (*p* < 0.05).

**Figure 10 ijms-24-16807-f010:**
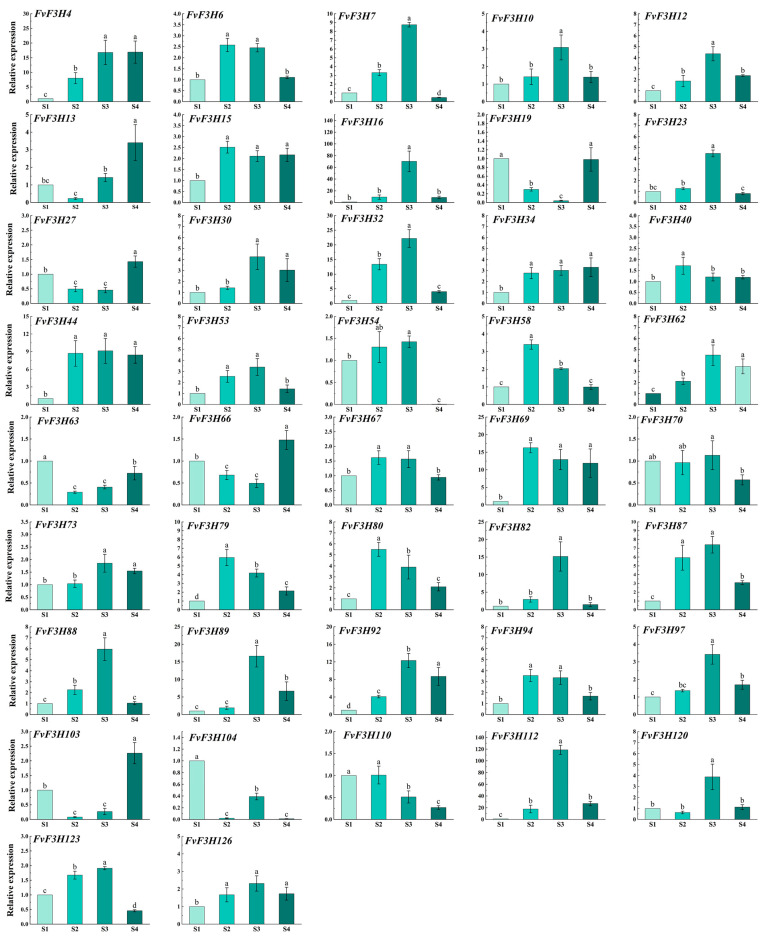
Relative expression levels of *F3H* gene in strawberry treated at different periods. S1 period was used as control. The 2^−∆∆*Ct*^ method was used to calculate the relative expression. Error bars represent the mean ± SE from three biological repeats. Different letters denote significant differences (*p* < 0.05), whereas the same lowercase letters indicate no statistical difference.

## Data Availability

Data will be made available on request.

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
