# Peer review of "Identification of Flavanone 3-Hydroxylase Gene Family in Strawberry and Expression Analysis of Fruit at Different Coloring Stages"

_ijms, 2023, doi:10.3390/ijms242316807_

Round 1
Reviewer 1 Report
Comments and Suggestions for Authors
The article on the surface looks good unfortunately I have the impression that the authors do not understand the concept of a gene. This article has serious flaw for example:
Abstract: "Secondary structure prediction indicated that the gene family was dominated by random coils and α-helices, mainly located in chloroplasts, cytoplasm, nucleus and cytoskeleton."
Results: "Subcellular localization prediction results showed that F3H gene family members were mainly located in chloroplasts, cytoplasm, nucleus and cytoskeleton, 29 genes were located in mitochondria, 21 genes were located in vacuoles, 22 genes were located in Golgi bodies, and only 14 genes were located in endoplasmic reticulum."
Discussion: "found that most FvF3H family members were located in chloroplasts, cytoplasm, nucleus and cytoskeleton, while a few were located in endoplasmic reticulum, mitochondria, Golgi and vacuole"
Conclusion: "FvF3H genes were mainly located in chloroplasts, cytoplasm, nucleus and cytoskeleton"
This is the first time I've seen that vacuoles, Golgi bodies, cytoskeleton and reticulum contain genes!!!! Genes may be responsible for the function of these organelles but these organelles do not contain genes!
There are also small mistakes in case latin names, e.g., Dianthuscaryophyllus, Antirrhinummajus,
Reviewer 2 Report
Comments and Suggestions for Authors
This manuscript by Yanqi Zhang and colleagues covers an interesting topic related with anthocyanin biosynthesis in strawberries. In general, the research work is correct and understanding to the readers. Authors show a great number of valuable results. However, in my opinion, the work could be improve introducing a few changes in several points before a possible publication.
1 - The abstract is so much complex. I suggest only to introduce the main results. In addition, a sentence related with the background of the topic studied could be useful to demonstrate the state of art about the topic studied and how this study can be innovative.
2 - Line 88-90: this sentence could be introduced in the conclusions.
3 - Line 234: “… color transformation.” In my opinion, this is not a correct way to define the color changes during the strawberry maturation and their relationship with anthocyanins biosynthesis. Probably: strawberry anthocyanin accumulation or biosynthesis.
4 - It is necessary for the authors to consider that during the biosynthesis of these pigments, there is no accumulation of a single anthocyanin, but rather a sequence of different anthocyanins forms that are biosynthesized. The way the authors write gives a vision that is just a form. I suggest introducing this topic.
5 - Figure 9: legend related with the statistical analysis is missing. So, “Different letters ….”
6 - Line 365-368: Name of the strawberry variety studied is missing. It is not totally clear how was carry out the sampling and its size.
7 - Line 421-431: reference for extraction conditions and anthocyanin determination are missing. Extractions were performed in duplicate or triplicate. this information is missing. The authors indicate that the determination and acquisition of readings on the spectrophotometer were carried out in triplicate. But this is just a triplicate of the determination and not of the extraction process. Clarify this topic.
8 - Line 424: “shaken several times”. This is not clear. Duration and shake rotations are missing.
9 - Line 429-431: What calibration curve was performed? Which standard anthocyanin was used? These data are missing.
10 - Line 453: In my opinion, its more correct to describe “anthocyanins” (in plural).
11 - 458: “… strawberry coloration.” Suggest changing for “… strawberry color changes”.
Round 2
Reviewer 1 Report
Comments and Suggestions for Authors
Dear Authors,
I am sorry but still paper needs serious correction.
In eukaryotes, the transcription process of nuclear genes takes place in the cell nucleus, while the translation process takes place in the cytosol. Thus, your changes contain serious mistakes e.g.,: "Subcellular localization prediction results showed.The F3H gene family is mainly expressed in cytoplasm, chloroplasts,Nucleus, mitochondria and cytoskeleton"
Round 3
Reviewer 1 Report
Comments and Suggestions for Authors
The article needs a thorough reading by the authors and thorough revisions. Minor errors, show the authors' lack of respect for both Editors and reviewers. Please do a language correction.
Please do not use English names of plants but Latin names. Please correct throughout the text.
Can the authors explain to me what plant they mean when they use the term "selenium"?
Is this sentence correct: "This study will provide a way to further understand the role of F3H gene family in strawberry color changes and laid a foundation for further study of 479 biological func-tions and molecular mechanisms of F3H gene members in strawberry gene members in strawberry."
Are proteins located or connected with "and cytoskeleton, and a few were located in the endoplasmic reticulum, mitochondria, Golgi apparatus and vacuoles"?
